# Prenatal Risk Factors for Brief Resolved Unexplained Events in Infants

**DOI:** 10.3390/pediatric17010016

**Published:** 2025-02-06

**Authors:** Luana Nosetti, Marco Zaffanello, Eliot S. Katz, Elisa Morrone, Michele Abramo, Francesca Brambilla, Antonella Cromi, Giorgio Piacentini, Massimo Agosti

**Affiliations:** 1Pediatric Sleep Disorders Center, Division of Pediatrics, “F. Del Ponte” Hospital, University of Insubria, 21100 Varese, Italy; luana.nosetti@uninsubria.it (L.N.); mabramo1@studenti.uninsubria.it (M.A.); fbrambilla5@studenti.uninsubria.it (F.B.); 2Department of Surgery, Dentistry, Pediatrics and Gynecology, University of Verona, 37126 Verona, Italy; marco.zaffanello@univr.it (M.Z.); giorgio.piacentini@univr.it (G.P.); 3Division of Sleep Medicine, Johns Hopkins All Children’s Hospital, St. Petersburg, FL 33701, USA; 4IRCCS Humanitas Research Hospital Rozzano, 20089 Milano, Italy; elisa.morrone@mc.humanitas.it; 5Department of Obstetrics and Gynecology, University of Insubria, 21100 Varese, Italy; antonella.cromi@uninsubria.it; 6Department of Medicine and Surgery, University of Insubria, 21100 Varese, Italy; massimo.agosti@uninsubria.it

**Keywords:** apparent life-threatening event, brief resolved unexplained events, sudden infant death syndrome

## Abstract

Background: Prenatal factors have been implicated in the likelihood of reporting sleep disorders in infants. The influence of prenatal and pregnancy-related factors on the incidence of brief resolved unexplained events (BRUEs) in infants has not been established. Objectives: This study aims to evaluate the prenatal and pregnancy-related factors that may contribute to the development of BRUEs in infants. Methods: A single-center, observational, and cross-sectional cohort study was conducted on mothers of children presenting to the Pediatric Clinic of the University of Insubria’s Center for the Study of Respiratory Sleep Disorders with BRUEs as infants. The mothers of typically developing children were enrolled as a control group consecutively at their respective outpatient clinics. All mothers were administered comprehensive questionnaires including demographics, past medical histories, and pregnancy-related issues (weight gain, Berlin sleep-disordered breathing score, and insomnia severity index), psychological symptoms, medical history, illnesses, and medications. Results: Infants with BRUEs were delivered at an earlier gestational age. Mothers of infants with BRUEs were more likely to snore during pregnancy and have lower extremity edema during the first trimester, uterine contractions and restless legs syndrome symptoms during the second trimester, and muscle aches and aspirin usage during the third trimester. The insomnia severity index composite score was not different between the control and BRUE groups. Mothers of infants with BRUEs were less likely to report leg cramps, pregnancy-related diarrhea, fatigue, and gastroesophageal reflux. Conclusions: Mothers of infants presenting with BRUEs had more symptoms during pregnancy of snoring and uterine contractions but not insomnia and were less likely to report leg cramps, pregnancy-related diarrhea, fatigue, and gastroesophageal reflux. The reporting of this study conforms with the STROBE statement.

## 1. Introduction

The nomenclature regarding infants presenting with paroxysmal episodes of changes in tone, breathing, or color has evolved to promote specificity in diagnosis, limit unnecessary testing, enhance prognostication, and facilitate future research. The original terminology, “near-miss sudden infant death syndrome (SIDS)”, erroneously conflated these events with a fatal illness of incompletely understood pathophysiology. In 1986, another term was introduced, “apparent life-threatening event (ALTE)”, which was similarly problematic as the risk of severe illness in these infants was less than 5% [1]. In 2016, an American Academy of Pediatrics expert committee coined the term, “brief resolved unexplained events (BRUEs)”, to describe a portion of infants presenting with paroxysmal events that were brief (<1 min), had resolved by the time of presentation, and for whom no diagnosis was evident after a comprehensive clinical history and physical examination [2].

A major limitation to accurately characterizing a BRUE is that caregivers may have markedly disparate thresholds for reporting concerning behavior in their infants. When mothers of infants with sudden infant death syndrome (SIDS) were asked about the presence of cyanosis, apnea, wheezing, and unusual breathing patterns, there were significant differences between mothers employed as nurses and non-nurse mothers indicating that experience may modify the perception of infant behavior [3]. Similarly, there are reports that prenatal psychological traits in the parents are predictive of which infants will be reported to have a sleep disorder [4,5]. However, few data have been presented on prenatal and pregnancy-related factors of mothers, at any point before or during the pregnancy, with and without infants reported to have experienced BRUEs.

The incidence of ALTEs or BRUEs has been estimated from hospitalizations, emergency room visits, and retrospective questionnaires. These studies suggest that between 3 and 5% of parents report a stoppage of breathing or apnea in their infant [6] though only 0.5% are brought in acutely for medical evaluation, and only 0.9% are admitted to the hospital for evaluation [6]. Further, the collaborative home infant monitoring study (CHIME study) prospectively evaluated infants with documented monitoring and found apneic pauses exceeding 20 s in 40% of infants, periodic breathing during sleep in most infants, and that term infants spend an average of 6 s per hour with an oxygen saturation below 90% [7,8]. Thus, respiratory pauses, irregular breathing, and hypoxemia are very common in infants yet only a fraction of infants are reported to have BRUEs. When asked, “Did baby ever stop breathing for more than 20 s”, 5.3% of parents answered affirmatively [6]. The reporting of infant apnea was increased in maternal smokers, infants who had a low birthweight or short duration, and those that were admitted to the neonatal intensive care unit [6].

Pregnancy is characterized by anatomical, hormonal, biochemical, and psychological changes that can negatively impact sleep quality [9]. This study aims to identify maternal health conditions and pregnancy-related factors that may alter the risk of BRUEs in infants. The findings of this study could have significant implications for clinical practice, helping to inform prevention and management strategies for BRUE episodes for at-risk infants.

## 2. Methods

This was a single-center, observational, and cross-sectional cohort study conducted from November 2020 to April 2021. The mothers of children presenting to the Pediatric Clinic of the University of Insubria’s Center for the Study of Respiratory Sleep Disorders with brief resolved unexplained events (BRUEs) were enrolled. Inclusion criteria included mothers of all typically developing infants who experienced BRUEs. Exclusion criteria included mothers of children with genetic disorders or other major cardiac, respiratory, or neurological conditions. The mothers of typically developing children born at Filippo del Ponte Hospital ASST Sette Laghi di Varese were also enrolled as a control group. Participants were enrolled consecutively to reduce bias at their respective outpatient clinics. Inclusion criteria included mothers of all typically developing infants who had never experienced BRUEs. Exclusion criteria included mothers of children with genetic disorders or other major cardiac, respiratory, or neurological conditions. All patients agreed to participate and signed an informed consent form. The study was approved by the Ethics Committee at the University of Insubria (protocol number 05092017). The reporting of this study conforms with the STROBE statement.

All mothers were administered a comprehensive questionnaire (see Appendix B) including demographics, past medical history, and pregnancy-related issues (weight gain, Berlin sleep-disordered breathing score, National Institute for Health and Care Excellence (NICE, 2014), psychological symptoms (during the last month of pregnancy), medical history, illnesses, medications, and insomnia severity index). The Berlin questionnaire was scored as “high-risk” or “low-risk” for obstructive sleep apnea using standard techniques [10]. The insomnia severity index pertained to the third trimester and was calculated by assigning a total score between 0 and 28. The score is interpreted as follows: absence of sleep disturbance (score 0–7), sub-threshold or subclinical insomnia (score 8–14), moderate insomnia (score 15–21), and severe insomnia (score 22–28) [11,12].

### Statistical Analysis

Data analysis was performed using SPSS version 22.0 for Windows (SPSS Inc., Chicago, IL, USA). Continuous variables with a normal distribution are expressed as the mean (standard deviation), while those without a normal distribution are defined as the median (M) and interquartile range (IQR). Categorical variables are reported as absolute percentages (numbers). Differences between continuous variables between groups were assessed using Student’s *t*-test or Mann–Whitney’s U-test, and differences in the distributions of categorical variables were evaluated using either the chi-squared test or Fisher’s exact test, whichever was appropriate. A binary logistic regression was performed considering ALTEs/BRUEs in infants as a dichotomous variable (no = 0; yes = 1). The regression was used to identify maternal factors independently associated with the risk of ALTEs/BRUEs in infants. The main maternal confounding factors were considered in the binary logistic regression analysis. All tests were two-sided, and a *p*-value < 0.05 was considered statistically significant. There were no missing data.

## 3. Results

One hundred and eleven mothers of healthy children were invited to participate in the study as a control group, and 110 of these mothers agreed. The mother who chose not to participate reported that she did not have the time that day to complete the survey. Eighty-eight mothers of infants with a history of BRUEs were all invited and agreed to participate in the study. Thus, the study group included 88 mothers of children with a history of BRUEs (BRUE mothers) and 110 mothers of healthy, control children (control mothers) without a BRUE history.

Table 1: The mothers’ age and weight changes during pregnancy were similar. The BRUE group was born at an earlier gestational age. No significant differences were found in pre-existing maternal medical conditions.

Table 2: Compared to control mothers, BRUE mothers more often reported snoring in the first trimester, but did not otherwise differ in their degree of snoring or the Berlin questionnaire.

Table 3: Compared to control mothers, BRUE mothers were (1) more likely to report lower extremity edema during the first trimester; (2) more likely to have uterine contractions and restless legs syndrome symptoms during the second trimester; and (3) more likely to have muscle aches and aspirin usage, but less likely to have leg cramps, gastroesophageal reflux, diarrhea, and tiredness during the third trimester.

Table 4: No significant differences were found in psychological symptoms between mothers with and without an infant experiencing a BRUE.

The insomnia severity index composite score was not different between the control and BRUE groups (8.9 ± 5.8 vs. 8.7 ± 6.7). In evaluating medication usage during pregnancy, only aspirin use in the third trimester was increased in BRUE mothers (see online repository for Appendix A for all medication usage).

The binary logistic regression analysis demonstrated that snoring in the first trimester, leg movements, uterine contractions during the second trimester, and snoring and body aches during the third trimester, as well as preterm birth, are independent risk factors for BRUEs in offspring. By contrast, leg cramps, gastroesophageal reflux (GER), fatigue, and diarrhea during the third trimester appear to be protective factors against BRUEs in offspring (see online repository for Appendix A).

## 4. Discussion

This is the first large study to comprehensively evaluate prenatal and pregnancy-related factors as a determinant of BRUE presentations in infancy. Infants with BRUEs had a lower gestation age compared to control infants without BRUEs. During pregnancy, mothers of infants with BRUEs more often reported snoring in the first trimester, having leg cramps, and having lower extremity edema. During the second trimester, mothers of infants with BRUEs were more likely to have uterine contractions and restless legs symptoms. Finally, during the third trimester, mothers of infants with BRUEs were more likely to have muscle aches and more aspirin usage but less diarrhea, fatigue, and GER. There were no differences between the two mother groups in relation to insomnia or psychiatric symptoms. The binary logistic regression analysis indicated that leg movements and uterine contractions during the second trimester, as well as snoring during the first and third trimester, body aches in the third trimester, and earlier gestational age are risk factors for BRUEs in offspring.

### 4.1. Relation to SIDS

Scoring systems with prenatal and post-natal risk factors have been developed to identify infants at risk for SIDS so that resources could be directed towards prevention interventions [13]. Given that prenatal parent psychological traits are predictive of infants reporting to have a sleep disorder [4,5], and that a variety of irregular breathing patterns are normally observed in infants [7,8], it is plausible that educational interventions directed towards at-risk parents could mitigate the perception that an infant has experienced a BRUE. In a cohort of nurses who were mothers of infants with SIDS, 37% reported a prior history of cyanosis, apnea, wheezing, or unusual breathing patterns, compared to only 6% of non-nurse mothers of SIDS infants [3]. A control group of nurses without a SIDS infant did not report these abnormal breathing patterns. Thus, it is plausible that parents with specific training may be more likely to identify abnormal breathing patterns in their infants. Extending this current analysis to larger studies may provide the basis for developing prevention programs for infants at risk for BRUE analogous to programs targeting SIDS [14].

The epidemiology of SIDS and ALTEs has been extensively studied, but there are no data on the epidemiology of BRUEs. In the United States, in 2020, the sudden unexplained infant death (SUID) rate was 0.00093% while the SIDS mortality rate was 0.00038% of live births [15]. By contrast, the incidence of ALTEs in population-based studies has been reported in 0.5% of infants. The age of death for SIDS cases is typically 2–3 months old [16,17] compared to 2 weeks old for the presentation of BRUE cases. The maternal and pregnancy-related factors that influence SIDS include gestational age, obtaining prenatal care, maternal age, maternal smoking, alcohol use, and illicit drugs [18]. In our study, maternal age, maternal smoking, and substance abuse were not identified as risk factors for BRUEs, but lower gestational age was predictive of an infant’s likelihood of having a BRUE. Preterm birth has previously been identified as a risk factor for both BRUEs and ALTEs [19,20]. It is not surprising that preterm birth is a risk factor for SIDS, ALTEs, and BRUEs, as respiratory immaturity is well documented at earlier gestational ages, increasing the risk of apnea, periodic breathing, and hypoxemia [21].

SIDS infants are more likely to be male and to be born to younger mothers [17]. SIDS cases are also more common in lower standard-of-living households, single-parent families, infants with low birthweight [22], and those with less post-natal infant healthcare [17]. By contrast, the ALTE literature indicates that there is no sex predisposition and that the age of the mother is not a risk factor [20]. SIDS is reported to be higher in the winter months [17], but no such seasonality is observed with BRUEs. Our results are consistent with the ALTE literature insofar as maternal age and male sex were not predictive of having a BRUE. Overall, although there is some overlap, SIDS and BRUEs appear to have separate epidemiological risk factors and are therefore distinct entities.

Maternal smoking during pregnancy can have adverse effects on fetal health and lead to complications, such as decreased blood flow and lower birthweight [23,24]. It is also a risk factor for SIDS [25]. In our study, there was no significant difference in smoking status during pregnancy between mothers of children with BRUEs and mothers of healthy children.

### 4.2. Maternal Weight

Infants born to pregnant women who are obese, defined as having a pre-pregnancy BMI of 30 kg/m^2^, have a higher risk of neonatal death [26]. However, the link between obesity and perinatal death may be mediated by the gestational age at delivery [27]. Many other factors can impact the risk of neonatal death [28,29]. In our study, we found no statistically significant differences in BMI (kg/m^2^) prior to pregnancy between the groups, indicating that this variable did not significantly impact BRUEs. Weight gain beyond the recommended amount appears protective against infant mortality [30]. In our study, however, pregnancy weight gain did not appear to be a risk factor or protective for BRUEs.

### 4.3. Maternal Sleep Disorders

Pregnancy is characterized by anatomical, hormonal, biochemical, and psychological changes that can negatively impact sleep quality [9]. Sleep disturbances are common during pregnancy but often overlooked in clinical practice [31]. It is crucial to recognize them because they can affect the mother and fetal health. Several factors, including psychological stress related to pregnancy, physical discomforts such as finding a comfortable position, nocturia, and restless legs syndrome, can contribute to insomnia [32]. Insomnia during pregnancy has been linked with an increased risk of postpartum depression which could reasonably adversely affect care-giving and the perception of infant illnesses [33].

Pregnancy-related problems such as difficulty falling asleep and/or staying asleep, impact the quality of life and increase the risk of preterm birth and postpartum depression [34]. Previous studies have reported that these sleep disorders present and/or worsen as the pregnancy progresses [9,32]. Maternal insomnia during pregnancy can negatively affect the mother’s overall health, fetal growth, adverse perinatal outcomes, and increase the risk of SIDS [35]. Thus, it is plausible that maternal insomnia during pregnancy may increase the vulnerability of infants to BRUEs. Lack of adequate sleep can impair a mother’s ability to stay alert during the day and care for her baby. However, in our study, the “insomnia severity index” questionnaire was similar between the two groups.

We evaluated whether snoring during pregnancy is a risk factor for BRUEs. Maternal snoring can interfere with the mother’s sleep quality and increase the risk of sleep apnea, which can affect the fetus’s oxygenation [35,36]. Snoring and sleep apnea increase the risk of complications for both mother and fetus [37,38]. In our study, mothers of children with BRUEs reported snoring slightly more frequently during the first trimester of pregnancy than mothers in the control group. However, the study found no statistically significant differences in the intensity of snoring, the ability to disturb others, and the presence of apnea between mothers of affected children and mothers of control children. This finding suggests that there may not be a significant difference in snoring characteristics between the two groups, but more research is needed.

Restless legs syndrome (RLS) is a common sleep disorder that can occur during pregnancy [39]. The symptoms of RLS increase as pregnancy progresses, reaching a peak in the third trimester [40]. One study showed that leg movements such as hopping or jerking, which are representative symptoms of RLS, were associated with a higher prevalence of earlier gestational age and lower birthweight [41]. Another study found that gestational RLS was not linked to fetal distress or low Apgar scores [42]. Our study showed that mothers of children with BRUE episodes had increased second-trimester RLS compared to mothers in the control group.

Nocturnal leg cramps differ from restless legs syndrome, which is characterized by an involuntary movement of the legs without muscle contractions or pain. Both conditions can occur in pregnant women, with up to 30–50% of pregnant women experiencing leg cramps, especially in the third trimester. Leg cramps during pregnancy can lead to difficulties during labor, fetal hypoxia, and an increased risk of neonatal asphyxia [43]. In our study, nocturnal leg cramps in the first trimester were statistically more significant in the control group and less in the BRUE group.

Nocturnal uterine contractions are common during pregnancy and often do not indicate a problem. However, having uterine contractions early in pregnancy could be a factor in an increased risk of preterm delivery. In our study, nocturnal vigorous uterine contractions in the second trimester were higher in the BRUE group.

### 4.4. Parental Psychological Traits

Parental psychological traits, including anxiety or depression, which were present long before their child’s birth, are highly predictive of parents who will report that their infants have sleep disturbances [5,44]. Post-natal depression is also a recognized risk factor for SIDS [45]. Depression and anxiety can impair a mother’s ability to stay alert and care for her baby during the day. Additionally, depression and anxiety can negatively affect the mother’s overall health and fetal outcomes [46,47,48]. Since the reporting of BRUEs is subjective, it is reasonable to evaluate whether parental psychological traits influence BRUE presentations. However, in our data, there were no significant differences between the maternal groups with regard to pregnancy-related anxiety or depression.

### 4.5. Maternal Illnesses

We investigated whether pre-existing maternal illnesses could increase the risk of BRUEs in infants. For instance, cardiovascular, metabolic, or respiratory diseases can impact the fetus during pregnancy and birth. Our study analyzed maternal pathologies before pregnancy, such as arterial hypertension, pre-gestational diabetes, gastroesophageal reflux (GER), and cardiovascular, pulmonary, and metabolic pathologies. However, these were not found to be significantly associated with BRUEs.

Sleep quality is worse in pregnant mothers with GER, particularly in obese mothers [49]. The prevalence of GER is high during pregnancy, often in the second and third trimesters [50]. However, the relationship between GER in pregnant mothers and BRUEs in infants has not been reported. In our study, GER was most frequent during the first trimester of pregnancy in mothers in the control group and least frequent during the third trimester of pregnancy in mothers whose children had BRUEs.

There is no direct relationship between lower extremity edema during pregnancy and the development of BRUEs in infants. However, lower extremity edema can signify an underlying condition such as preeclampsia, increasing the risk of complications during pregnancy and after delivery. Perinatal death and severe neonatal morbidity have been reported to be high among women with preeclampsia and eclampsia [51]. Indeed, maternal hypertensive disorders, particularly eclampsia and severe preeclampsia, have been associated with preterm birth, low birthweight, and increased risk of all-cause infant mortality [52,53,54]. In our study, no difference was found between hypertension during pregnancy in mothers of healthy children and those whose children developed BRUEs.

Maternal fatigue can hinder a mother’s ability to get enough sleep and stay awake during the day. Over 90% of pregnant women experience fatigue in the first, second, and third trimesters of pregnancy. Studies have shown a significant link between severe fatigue and a higher rate of preterm delivery and low birthweight [55]. In our study, mothers of children with BRUEs had a lower rate of fatigue during the third trimester of gestation.

Many pregnant women experience musculoskeletal pain and symptoms, particularly in the third trimester [56]. Pregnancy-related hormonal and physiological changes increase the risk of musculoskeletal problems during pregnancy, which can complicate pregnancy, especially in the third trimester. Though there is no established connection between muscle pain during pregnancy and the development of BRUE in the infant, in our study, mothers with muscle aches were more likely to have children who developed BRUEs.

Medication use during pregnancy must balance the benefit to the mother and the potential harm to fetal growth and development. Exposure to any addictive drug during pregnancy increases the risk of SIDS [57]. Certain medications, such as benzodiazepines, opioids, tricyclic antidepressants, and anticonvulsants, can increase the risk of medical illness in the newborn if taken during pregnancy [58]. In our study, there was no significant difference in the use of antibiotics, paracetamol, antidepressants, anxiolytics, anti-inflammatories, heparin, and levothyroxine between the two groups. Only aspirin was used more frequently in the third trimester of pregnancy by mothers of children who experienced BRUEs. Aspirin use during pregnancy reduces the risk of maternal hypertensive disorders, and early use in high-risk women is associated with a lower incidence of preeclampsia/eclampsia [59]. Low-dose aspirin prophylaxis in women at risk of preeclampsia may also reduce the risk of fetal growth delays [60].

### 4.6. Limitations

Though BRUE mothers and controls were queried during the same calendar year, the BRUE mothers reported a pregnancy that was further in the past. Thus, there may be a recollection bias in our data. The study group of BRUE mothers all came to the outpatient clinic for evaluation or follow-up which may rule out mothers with certain psychological traits or socio-economic factors. It is also possible that the sample size is insufficient to detect a significant difference. This study was conducted during the COVID-19 pandemic which was associated with a variety of respiratory illnesses and sleep disorders that may have altered the mother’s recollection of events during pregnancy. Nevertheless, the patients and controls were recruited consecutively and are therefore likely to be representative of the cases and controls in the community. Marital status and socio-economic data were not obtained but may be important determinants of the occurrence of BRUEs. This was a single-center study, and therefore, the findings may not be generalizable to populations with different socio-economic or medical conditions.

## 5. Conclusions

Maternal insomnia, anxiety, depression, and stress during pregnancy were not found to be significant risk factors for BRUEs. However, in the mothers of infants with BRUE cases, snoring and body aches in the third trimester and vigorous nocturnal uterine contraction and restless legs syndrome in the second trimester were significantly more common. In addition, mothers of infants with BRUE cases were less likely to have reported diarrhea, fatigue, and GER in the third trimester.

## Figures and Tables

**Table 1 pediatrrep-17-00016-t001:** Demographics, pregnancy weights, and pre-pregnancy diseases.

		Control Group	BRUE Group	
Sample size		110	88	
	Variable			*p*-value
Infant	Sex (% males)	43.6	51.1	0.182
	Gestational age (wks)	39.1 ± 1.3	38.3 ± 2.1	0.001 *
Maternal	Age (years)	32.6 ± 5.4	33.7 ± 5.2	0.162
	Pre-pregnancy weight (kg)	62.0 ± 12.2	61.5 ± 11.8	0.758
	Height (m)	1.64 ± 0.06	1.64 ± 0.06	0.788
	Pre-pregnancy BMI	23.1 ± 4.3	22.9 ± 4.1	0.643
	Weight at delivery (kg)	74.1 ± 11.3	73.3 ± 13.0	0.637
	Pregnancy weight gain (%)	20.5 ± 8.7	19.8 ± 8.8	0.545
Race (n)	Caucasian	105	75	0.156
	Black	1	3	
	Asian	3	6	
	Hispanic	1	3	
	Other	0	1	
Prior *	Hypertension (%)	3.6	3.4	0.622
	Gastroesophageal reflux (%)	14.5	14.8	0.560
	Cardiovascular disease (%)	3.6	3.4	0.622
	Lung diseases (%)	0	3.5	0.086
	Metabolic diseases (%)	0.9	2.3	0.417
	Diabetes (%)	2.7	1.1	0.399

Chi-squared or two-tailed Student’s *t*-tests. Continuous variables are mean ± SD. * *p* > 0.05 = comes from the reference distribution.

**Table 2 pediatrrep-17-00016-t002:** Snoring during pregnancy.

	Trimester	Answer	Controls (n = 110)	BRUE (n = 88)	*p*-Value
**Did you snore at night during pregnancy? (%)**	1st	Yes	10.9	22.7	
		No	74.5	69.3	
		I don’t know	14.5	8.0	0.044 *
	2nd	Yes	23.6	26.1	
		No	63.6	64.8	
		I don’t know	12.7	9.1	0.699
	3rd	Yes	40.9	31.8	
		No	47.3	59.1	
		I don’t know	11.8	9.1	0.254
**Quality of snoring if present (%)**	1st	1 = Slightly louder than a breath	50	55	
		2 = Loud as talking	42	35	
		3 = More noisy than talking	0	0	
		4 = Very noisy	8	10	0.929
	2nd	1	46	43	
		2	34	43	
		3	11	4	
		4	8	9	0.788
	3rd	1	47	43	
		2	31	29	
		3	11	14	
		4	11	14	0.945
**Frequency of snoring (%)**	1st	1 = More or less every night	75	45	
		2 = 3–4 nights a week	25	10	
		3 = 3–4 nights per month	0	30	
		4 = 1–2 nights a month	0	15	0.053
	2nd	1	50	43	
		2	27	13	
		3	15	30	
		4	8	13	0.418
	3rd	1	42	36	
		2	22	18	
		3	9	32	
		4	27	14	0.079
**Has your snoring ever bothered other people? (%)**	1st	Yes	50	35	
		No	50	65	0.320
	2nd	Yes	42	48	
		No	58	52	0.460
	3rd	Yes	47	54	
		No	53	46	-
**Has anyone noticed that you stopped breathing while sleeping? (%)**	1st	1 = More or less every night	0	10	
		2 = 3–4 nights a week	0	0	
		3 = 1–2 nights a week	0	0	
		4 = 3–4 nights a month	8	0	
		5 = 1–2 nights a month	92	90	0.238
	2nd	1	0	9	
		2	4	0	
		3	0	0	
		4	12	0	
		5	85	91	0.119
	3rd	1	0	7	
		2	2	0	
		3	0	0	
		4	9	7	
		5	89	86	0.270
**Berlin Questionnaire (%)**	1st	Negative	92	80	
		Positive	8	20	0.366 (Fisher)
	2nd	Negative	92	83	
		Positive	8	17	0.276 (Fisher)
	3rd	Negative	93	82	
		Positive	7	18	0.136 (Fisher)

* *p* < 0.05.

**Table 3 pediatrrep-17-00016-t003:** Diseases and symptoms during pregnancy.

	Trimester of Pregnancy	Controls (n = 110)	Cases (n = 88)	*p*-Value
GER (%)	1	22.7	14.8	0.109
	2	29	19	0.077
	3	51	33	0.006 *
Gestational diabetes (%)	1	0.9	2.3	0.417
	2	7.3	11.4	0.227
	3	10.9	18.2	0.105
Hypertension (%)	1	0.9	3.4	0.232
	2	0.9	5.7	0.063
	3	9	11.4	0.384
Edema in lower limbs (%)	1	0	5.7	0.016 *
	2	5.4	12.5	0.067
	3	27.3	28.4	0.492
Irrepressible feeling of moving your legs (%)	1	0.9	4.5	0.123
	2	3.6	12.5	0.019 *
	3	11.8	14.8	0.343
Nocturnal cramps in lower extremities (%)	1	13.6	3.6	0.010 *
	2	17.3	26.1	0.090
	3	42.7	37.5	0.275
Vigorous nocturnal uterine contractions (%)	1	0	2.3	0.196
	2	0.9	8.0	0.015 *
	3	18.2	19.3	0.240
Need to urinate several times during night (%)	1	25.5	34.1	0.121
	2	48.2	54.5	0.228
	3	80.0	72.7	0.150
Smoking during pregnancy (%)	1	12.7	9.1	0.282
	2	9.1	9.1	0.601
	3	8.2	9.1	0.508
Cold and nasal congestion	1	7.3	10.2	0.313
	2	16.4	10.2	0.149
	3	13.6	5.7	0.052
Sore throat	1	1.8	4.5	0.243
	2	5.5	1.1	0.104
	3	4.5	2.3	0.324
Fever	1	0	2.3	0.196
	2	0	0	-
	3	0.9	0	0.556
Cough	1	0	3.4	0.086
	2	0.9	1.1	0.693
	3	1.8	0	0.307
Muscle aches	1	0.9	5.7	0.063
	2	4.5	9.1	0.160
	3	5.5	15.9	0.014 *
Fatigue	1	33.6	38.6	0.281
	2	23.6	30.7	0.171
	3	58.2	43.2	0.025 *
Diarrhea	1	2.7	2.3	0.604
	2	3.6	1.1	0.262
	3	10.9	2.3	0.016 *
COVID-19 swab	1	0	1.1	0.444
	2	3.6	0	0.093
	3	0.9	2.3	0.417

* *p* < 0.05.

**Table 4 pediatrrep-17-00016-t004:** Psychological symptoms during pregnancy.

Question		Controls (n = 110)	Cases (n = 88)	*p*-Value
Have you felt most of the time during the last month and days down, depressed, or hopeless?	Last month	15	16	0.541
During the last approved month most of the time and days little interest or pleasure in getting things done?		21	20	0.541
During the last month have you felt most of the time and days nervous, anxious or restless?				
	never	26	34	
	a few days	54	41	
	>half of days	15	20	
	Every day	5	5	0.295
During the last month, have you been unable to stop or control your concerns?				
	never	55	61	
	a few days	35	26	
	>half of days	5	10	
	Every day	5	2	0.269

## Data Availability

The raw data supporting the conclusions of this article will be made available by the authors on request.

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
