# Peer review of "Prenatal Risk Factors for Brief Resolved Unexplained Events in Infants"

_pediatrrep, 2025, doi:10.3390/pediatric17010016_

Round 1
Reviewer 1 Report
Comments and Suggestions for Authors
The study aimed to examine the influence of prenatal and pregnancy-related factors on the incidence of Brief Resolved Unexplained Events (BRUE) in infants. This had not been discussed previously. The authors used a single-center, observational, cross-sectional cohort questionnaire-based methodology using mothers of children presenting to the Pediatric Clinic of the University of Insubria's Center for the Study of Respiratory Sleep Disorders with a BRUE as infants. The mothers of typically developing children were enrolled as a control group consecutively. All mothers were administered comprehensive questionnaires including demographics, past medical history and pregnancy-related issues (weight gain, Berlin sleep-disordered breathing score, insomnia severity index), psychological symptoms, medical history, illnesses, and medications. The findings showed infants with a BRUE were delivered at an earlier gestational age. Mothers of infants with a BRUE were more likely to snore during pregnancy, to report leg cramps and lower extremity oedema during the first trimester, to have uterine contractions and restless leg symptoms during the second trimester, and to have muscle aches, and aspirin usage during the third trimester (presumably for pre-eclampsia but differences in pre-eclampsia rates was insignificant). The insomnia severity index composite score did not differ between the control and BRUE groups. Mothers of infants with a BRUE were less likely to report pregnancy-related diarrhea, fatigue, and gastroesophageal reflux. The authors concluded that mothers of infants presenting with a BRUE had more symptoms during pregnancy of snoring, leg cramps, and uterine contractions but not insomnia and were less likely to report pregnancy-related diarrhea, fatigue, and gastroesophageal reflux. The study was conducted during the COVID-19 pandemic, but only a small number of cases and controls were swab-positive. The reporting of this study conformed with the STROBE statement.
The paper is well written. The limitations of the study relate to recall issues in a questionnaire-based survey. The authors should check if they can capture documented pregnancy outcome data if available (e.g. infant’s mother is not in a relationship (i.e. divorced, separated, or never married), if the infant was not the firstborn and when the mother resided in a socioeconomically disadvantaged area) as described in this paper: doi.org/10.1007/s00431-012-1896-0 as these factors may have a possible link with BRUE.
Author Response
COMMENT: 1. The limitations of the study relate to recall issues in a questionnaire-based survey.
Response 1: This limitation appears in the Discussion paragraph on limitations (Page 12-13. Line 719-720).
COMMENT 2: The authors should check if they can capture; 2.1 documented pregnancy outcome data if available (e.g. infant’s mother is not in a relationship (i.e. divorced, separated, or never married), 2.2 if the infant was not the firstborn and 2.3 when the mother resided in a socioeconomically disadvantaged area.
Response 2: Unfortunately, this type of data was not collected. We have now added this as a Limitation to our study (Page 13. Line 727-728)
Reviewer 2 Report
Comments and Suggestions for Authors
This study explored pregnancy- related and mothers’ prenatal risk factors for BRUEs. Data relevant to the period preceding pregnancy, as well as to each trimester of pregnancy, were collected by means of questionnaires that were administered to mothers with children who had experienced a BRUE and to control mothers. Several differences between answers of BRUE and control mothers were found, which were interpreted as possible risk factors for BRUEs.
The text is a bit longer than really needed, but is clear and fluent. Interpretation of some data could be reconsidered.
1. The authors repeatedly write that the BRUE mothers snored more than the control mothers. 1) Page 4 line 143 and page 11 line 250: Here we can read that the BRUE mothers were more likely to snore, or reported snoring more frequently, than the control mothers in the first trimester. However, according to Table 2, that difference was due to the mothers snoring only few times a month, which were represented only in the BRUE group: these individuals can hardly be defined snorers; 2) Page 9 lines 169: Here we can read the BRUE mothers were more likely to snore, without any specification of the trimester; 3) Page 9, line 176 and page 13 line 344: Here we can read that snoring in the third trimester (if I understand correctly) was a risk factor for BRUEs (page 9), or that it was significantly more common in the BRUE mothers (page 113), but this is not supported by Table 2. Overall, table 2 indicates that in early pregnancy the BRUE mothers more often reported snoring than the control mothers, but many of them snored just occasionally and softly. Moreover, the table seems to show that frequency and quality of snoring had a greater progression throughout pregnancy in the control mothers. To conclude, in my opinion the role of snoring needs to be reconsidered.
2. It should be clarified when the questionnaires were administered (After a BRUE? At some time after delivery? At the same time for all trimesters or not?). Besides, It is necessary to indicate which period the ISI was relevant to.
3. Table 2. Here, some values are expressed as n, some as % and other values both as n and as %. It is necessary to use the same units for all values.
4. Abstract line 26 and 32, page 4 line 146 and page 9 line 169. Contrary to the text, Table 3 shows that leg cramps in the first trimester were more common among the control mothers.
5. Page 3, lines 106-114. References for questionnaires should be included in the reference list, not just recalled in parentheses here.
6. Ref. 8 should be corrected.
Author Response
COMMENT: 1. The authors repeatedly write that the BRUE mothers snored more than the control mothers …. that difference was due to the mothers snoring only few times a month, which were represented only in the BRUE group: these individuals can hardly be defined snorers;
Response 1: We acknowledge that the reported snoring in the BRUE mothers was characterized by occasional occurrences, as noted in Table 2. rather than the more frequent snoring that typically defines chronic snorers in the sleep-disordered breathing population. We did not intend to classify these individuals as chronic snorers but rather to indicate that snoring, even in its occasional form, was more frequent in the BRUE mothers in the first trimester. Thus, It is a fair point that the reported frequency of snoring was not very common in the BRUE mothers. Nevertheless, it was statistically significant. We have modified the text to reflect the fact the snoring differences were significant but mild. Page 4. Line 153, Page 11. Line 637 and Page 9. Line 553 have had the modifier “slightly” to more precisely describe our findings.
COMMENT 2: Page 9 lines 169: Here we can read the BRUE mothers were more likely to snore, without any specification of the trimester;
RESPONSE 2: The trimester was added on Page 9. Line 553 (new line number) along with the “slightly” modifier.
COMMENT 3: Page 9, line 176 and page 13 line 344: Here we can read that snoring in the third trimester (if I understand correctly) was a risk factor for BRUEs (page 9), or that it was significantly more common in the BRUE mothers (page 113), but this is not supported by Table 2.
RESPONSE 3: In the unadjusted analysis, snoring was more frequent in the BRUE mothers only during the 1st trimeter but in the Binary Regression modeling, snoring was more frequent in the BRUE mothers in both the 1st and 3rd trimesters. We had now made this distinction clear on Page 9. Line 543.
COMMENT 4: Moreover, the table seems to show that frequency and quality of snoring had a greater progression throughout pregnancy in the control mothers.
RESPONSE 4: The slightly increase in reported snoring in the CONTROL mothers did not reach statistical significance.
COMMENT 5: It should be clarified when the questionnaires were administered (After a BRUE? At some time after delivery?
RESPONSE 5: In the METHODS on Page 2. Line 100 we state the the BRUE mothers were recruited when they presented to the pediatric clinic and the Control mothers were recruited from the post-delivery clinic on Page 3. Line 108.
COMMENT 6: It is necessary to indicate which period the ISI was relevant to.
RESPONSE 6:The ISI data pertained to "pregnancy-related issues" (see Page 3. Line 118-121).
COMMENT 7: Table 2. Here, some values are expressed as n, some as % and other values both as n and as %. It is necessary to use the same units for all values.
RESPONSE 7: This was fixed for clarity with all values referring to percentages.
COMMENT 8: Abstract line 26 and 32, page 4 line 146 and page 9 line 169. Contrary to the text, Table 3 shows that leg cramps in the first trimester were more common among the control mothers.
RESPONSE 8: This was corrected on in The Abstract Page 1. Line 30 - 33.
COMMENT 9: Page 3, lines 106-114. References for questionnaires should be included in the reference list, not just recalled in parentheses here.
RESPONSE 9: This was corrected and now the questionnaire reference is cited along with the other references.
COMMENT 10: Ref. 8 should be corrected.
RESPONSE: Reference 8 is correct.
Reviewer 3 Report
Comments and Suggestions for Authors
Thank you for the opportunity to review this interesting article. I have no reservations about the methodology of the study ni the ethical issues . Below are a few comments.
Line 76 - Please provide the time interval in which parents report certain symptoms, whether it occurred once in their lives or recurring, over what period the symptoms occurred to clarify this
Line 90 Please consider changing the purpose of the of the analysis so that a specific thesis is not assumed. Instead of 'that may in- 91 crease the risk of BRUE in infants' you may want to specify whether it has an impact.
Line 97 Please provide clear criteria for the inclusion and exclusion from the study.
Line 141 Please indicate statistical statistical significance in all tables. The race of the patients should be separated in the table.
Line 330 Limitations should take into account any possible limitations of the study both due to the selection of the study group study group, as well as the adopted study methodology. This section should be expanded.
Line 340 The first sentence of the conclusions should be removed. Conclusion is not an abstract.
References are correctly selected and up-to-date.
Tables should be graphically refined. Materials placed in the appendix. Please check the text for language.
Author Response
COMMENT 1: Line 76 - Please provide the time interval in which parents report certain symptoms, whether it occurred once in their lives or recurring, over what period the symptoms occurred to clarify this.
RESPONSE 1: The responses of the parents in these previous studies pertain to time time interval of the pregnancy or prenatally at any point. This has been added to Line 77.
COMMENT 2: Line 90 Please consider changing the purpose of the of the analysis so that a specific thesis is not assumed. Instead of 'that may in- 91 crease the risk of BRUE in infants' you may want to specify whether it has an impact.
RESPONSE 2: We have now changed the aim to be more neutral on Page 2 Line 95 to state the that the analysis will consider whether the various factors “alter the risk of BRUE.”
COMMENT 3: Line 97 Please provide clear criteria for the inclusion and exclusion from the study.
RESPONSE 3: We have now modified the Methods to explicitly state the Inclusion and Exclusion criteria. Page 3 Line 105-113
COMMENT 4: Line 141 Please indicate statistical statistical significance in all tables. The race of the patients should be separated in the table.
RESPONSE 4: The Tables have been modified to indicate the statistically significant findings and the Race was separated.
COMMENT 5: Line 330 Limitations should take into account any possible limitations of the study both due to the selection of the study group study group, as well as the adopted study methodology. This section should be expanded.
RESPONSE 5: We have now expanded the Limitations section as requested.Page 12-13. Lines 719-731
COMMENT 6: Line 340 The first sentence of the conclusions should be removed. Conclusion is not an abstract.
RESPONSE 6: The sentence at Page 13 Line 733 was removed as requested.
COMMENT 7: Tables should be graphically refined.
RESPONSE 7: The Tables have been extensively modified.
Reviewer 4 Report
Comments and Suggestions for Authors
1)The study design is well-defined as a single-center, observational, cross-sectional cohort study. However, given the topic's clinical significance, it would be useful to justify why a single-center study was chosen and whether the findings can be generalized to other populations?
2)The authors' adherence to the STROBE statement is commendable. Including the STROBE checklist as a supplementary file for reviewers/readers would be helpful to ensure all guidelines were followed.
3)The rationale for choosing mothers of typically developing children as a control group is valid but should be elaborated upon. Were there any specific exclusion criteria for control group participants that might introduce bias?
4)Were any efforts made to ensure that the two groups (mothers of children with ALTE/BRUE and the control group) were matched for confounding variables (e.g., socio-economic status)?
5)The scoring system for the Insomnia Severity Index is clearly defined, but was there any assessment of maternal insomnia over time to validate its association with ALTE/BRUE?
6)The authros does not mention whether adjustments for confounding variables, such as maternal age, socioeconomic status, or comorbid conditions, were made in the binary logistic regression analysis.
7) While the authors state there was no missing data, describing how this was ensured (e.g., through follow-up or data imputation methods) would strengthen the reliability of the findings.
8) In discussion section, the authros mention of scoring systems for SIDS risk factors is interesting, but it would be beneficial to elaborate on whether such scoring systems could be adapted or have been tested for BRUE.
9)The authros reported the potential for extending this analysis to larger studies. Are there ongoing or planned studies to explore these findings in a multicenter or more diverse population?
Author Response
COMMENT 1: However, given the topic's clinical significance, it would be useful to justify why a single-center study was chosen and whether the findings can be generalized to other populations?
RESPONSE 1: We have added this criticism to our Limitations paragraph (Page 12. Line 718). We believe that this is a representative population as the participants were recruited sequentially with few refusals. The population was drawn from a particular socioeconomic group which may indeed limit the generalizability of our findings.
COMMENT 2: Including the STROBE checklist as a supplementary file for reviewers/readers would be helpful to ensure all guidelines were followed.
RESPONSE 2: Our Strobe checklist was uploaded to the Supplemental Files section of the Journal.
COMMENT 3: The rationale for choosing mothers of typically developing children as a control group is valid but should be elaborated upon. Were there any specific exclusion criteria for control group participants that might introduce bias?
RESPONSE 3: We have now more explicitly stated our Inclusion and Exclusion criteria. There were no exclusions other than not having a typically developing child (Page 3. Line 105-113).
COMMENT 4: Were any efforts made to ensure that the two groups (mothers of children with ALTE/BRUE and the control group) were matched for confounding variables (e.g., socio-economic status)?
RESPONSE 4: We did not collect soccer-economic data on these patients. However, both the mothers in the Control group and in the BRUE group were derived from the same catchment area.
COMMENT 5: The scoring system for the Insomnia Severity Index is clearly defined, but was there any assessment of maternal insomnia over time to validate its association with ALTE/BRUE?
RESPONSE 5: The questionnaires from this study were obtained at a single time point and therefore longitudinal data are not available.
COMMENT 6: The authors does not mention whether adjustments for confounding variables, such as maternal age, socioeconomic status, or comorbid conditions, were made in the binary logistic regression analysis.
RESPONSE 6: In the binary logistic regression analysis, the confounding variables maternal age and co-morbid conditions (hypertension, diabetes, gastroesophageal reflux (GER), cardiovascular diseases, pulmonary diseases, and metabolic diseases) were added as covariates in the model (see the first and second columns, Supplemental Table 2). Each coefficient of the variables in the model represents the adjusted effect, namely the effect of the primary independent variable on the outcome, accounting for the other variables included in the model.
We added a statement in the Statistical Analysis section (Page 3. Line 137) specifying, "The main maternal confounding factors were considered in the binary logistic regression analysis.”
Socioeconomic status was not investigated (a study limitation acknowledged in line 351). In the Limitation section (line 352), we stated that "Marital status and socioeconomic data were not obtained but may be important determinants of the occurrence of a BRUE. This was a single-centre study; therefore, the findings may not be generalisable to populations with different socioeconomic or medical conditions."
COMMENT 7: While the authors state there was no missing data, describing how this was ensured (e.g., through follow-up or data imputation methods) would strengthen the reliability of the findings.
RESPONSE 7: Each questionnaire was checked for completeness at the time of completion leading to the very high data integrity achieved in this study.
COMMENT 8: In discussion section, the authors mention of scoring systems for SIDS risk factors is interesting, but it would be beneficial to elaborate on whether such scoring systems could be adapted or have been tested for BRUE.
RESPONSE 8: No scoring system has yet to be created for the prediction of BRUE, though this work and ensuing efforts may lead in that direction. We have stated this on Page 10 Line 578.
COMMENT 9: The authors reported the potential for extending this analysis to larger studies. Are there ongoing or planned studies to explore these findings in a multicenter or more diverse population?
RESPONSE 9: We are not currently engaged in any ongoing multi-center collaboration presently but we hope that the publication of this work will motivate consortiums and funding agencies to move forward with a larger prospective study with well-standardized, comprehensive pregnancy-related symptoms, soceo-economic data collection.
Reviewer 5 Report
Comments and Suggestions for Authors
I am grateful to the authors for the opportunity to read this interesting paper.
The authors have designed a cohort study, but I am particularly concerned that a sample size was not calculated prior to the development of the study. Was the necessary sample size calculated? If so, it should be explained how this calculation was made.
The authors indicate in the limitations that it is possible that the sample size was not sufficient to detect significant differences. This implies that the necessary sample size was not calculated before carrying out the study. This would not only be a limitation but a major methodological error.
The authors should explain these aspects very well and if the sample size was not calculated, they should justify why.
The tables should be homogenized. In the first table, the BRUE group appears in the left column and the control group on the right, and in the other tables the opposite. This may be a difficulty for the reader. The columns should be placed in the same order in all tables. In Table 1, I recommend removing the Kolmogoroc-S. column and changing the title of the last column to p-value (or similar). The "p" values should be marked with "*" or "+", etc., and the meaning (t-student, Chi-squarter) should be included at the bottom of the table. This comment is valid for all tables.
In the conclusion, the objective of the study should be answered with the results obtained. For this reason, the first two lines that talk about the type of study carried out are unnecessary. The last sentence should also be removed, which, although true, are interpretations of the authors that should be included, if considered necessary, in the discussion.
Author Response
COMMENTS:
COMMENT 1: Was the necessary sample size calculated? If so, it should be explained how this calculation was made.
RESPONSE 1: In our cross-sectional cohort observational study, we consecutively recruited mothers of children with a history of BRUE (88 mothers) and mothers of healthy children as a control group (110 mothers). The sample size was primarily determined by participant availability during the recruitment period and the study's available resources.
We did not perform an a priori sample size calculation based on specific statistical parameters because the study aimed to explore associations between maternal factors and BRUE events in a real clinical setting rather than test specific hypotheses with predefined power levels.
COMMENT 2: The tables should be homogenized.
RESPONSE 2: The Tables have been revised.
COMMENT 3: The objective of the study should be answered with the results obtained. For this reason, the first two lines that talk about the type of study carried out are unnecessary. The last sentence should also be removed, which, although true, are interpretations of the authors that should be included, if considered necessary, in the discussion.
RESPONSE 3: The requested changes have been made to the Conclusion on Page 13. Line 733.
Round 2
Reviewer 2 Report
Comments and Suggestions for Authors
Some points still require a refinement. Please, note that the line numbers I read in the revised version are not the same as those indicated in the replies.
Former point 1. I do not think writing “BRUE mothers were slightly more likely to snore” (page 4 line 148 and page 9, line 179) is an accurate description of the data yet. I suggest to write “BRUE mothers more often reported that in the first trimester they could happen to snore”. Besides, at page 9, line 169-171, the role of snoring in the third trimester as a risk factor should not be reported as a confirmation of the indications provided by the previous analysis, as it apparently contrasts with data shown in Table 2 (please, replace at least the word “confirms”).
Former point 2. The data about ISI, as well as those about psychological symptoms, are not reported in relation to each trimester. However, in the Appendix we can read that the ISI was administered in relation to the last trimester, and the psychological symptoms in relation to the last month of pregnancy. This should be clarified in the main text.
Former point 4. The result about leg cramps has been corrected in the abstract but not in the main text (pages 4 and 9).
Author Response
COMMENTS: 1. I do not think writing “BRUE mothers were slightly more likely to snore” (page 4 line 148 and page 9, line 179) is an accurate description of the data yet. I suggest to write “BRUE mothers more often reported that in the first trimester they could happen to snore”.
RESPONSE 1: Changes made on Page 5 Line 153 and Page 10. Line 184
COMMENT 2: at page 9, line 169-171, the role of snoring in the third trimester as a risk factor should not be reported as a confirmation of the indications provided by the previous analysis, as it apparently contrasts with data shown in Table 2 (please, replace at least the word “confirms”).
RESPONSE 2: The reviewer is corrrect that the word "comfirms" is not appropriated and has been replaced Page 10. Line 174.
COMMENT 3: The data about ISI, as well as those about psychological symptoms, are not reported in relation to each trimester. However, in the Appendix we can read that the ISI was administered in relation to the last trimester, and the psychological symptoms in relation to the last month of pregnancy. This should be clarified in the main text.
RESPONSE 3: The ISI did in fact refer to the third trimester and this has now been added to the Methods on Page 3. Line 122. The psychological symptoms did refer to the last month of pregnancy which has now been added to Page 3. Line 115.
COMMENT 4: The result about leg cramps has been corrected in the abstract but not in the main text (pages 4 and 9).
RESPONSE 4: Correction made page 5. Line 160 and Page 10. Line 178.
Reviewer 4 Report
Comments and Suggestions for Authors
Thank you !
Author Response
It looks to me like this reviewer had no additional Comments but simply checked the box to "not" sign the review report.